# Clinical Findings and Antimicrobial Susceptibility of Anaerobic Bacteria Isolated in Bloodstream Infections

**DOI:** 10.3390/antibiotics9060345

**Published:** 2020-06-19

**Authors:** Fernando Cobo, Jaime Borrego, Esther Gómez, Isabel Casanovas, Elizabeth Calatrava, Carla Foronda, José María Navarro-Marí

**Affiliations:** Department of Microbiology and Instituto de Investigación Biosanitaria ibs. GRANADA, University Hospital Virgen de las Nieves, 18014 Granada, Spain; jaime.borrego.sspa@juntadeandalucia.es (J.B.); esther.gomez.vicente.sspa@juntadeandalucia.es (E.G.); isabel.casanovas.sspa@juntadeandalucia.es (I.C.); elizabeth.calatrava.sspa@juntadeandalucia.es (E.C.); carla.foronda.sspa@juntadeandalucia.es (C.F.); josem.navarro.sspa@juntadeandalucia.es (J.M.N.-M.)

**Keywords:** bloodstream infections, antimicrobial susceptibility testing, MALDI-TOF MS, anaerobes

## Abstract

The objectives of this study were to report on the antimicrobial susceptibility of 141 clinically significant anaerobic bacteria isolated from bloodstream infections between January 2016 and April 2020 in a tertiary-care hospital in Granada (Spain) and to describe the main clinical features of the patients. Species identification was performed by MALDI-TOF MS (Bruker Daltonics, Billerica, MA, USA). Antimicrobial susceptibility tests were performed against penicillin, amoxicillin-clavulanic acid, imipenem, moxifloxacin, clindamycin, metronidazole, and piperacillin-tazobactam using the gradient diffusion technique and EUCAST breakpoints, except for moxifloxacin (CLSI breakpoints). The most frequent anaerobes were *Bacteroides* (43.9%, *n* = 62), *Clostridium* (24.1%, *n* = 34) and Gram-positive anaerobic cocci (GPACs) (15.6%, *n* = 22). Almost all tested anaerobes were susceptible to imipenem and amoxicillin-clavulanic acid, except for *Bacteroides*. High overall resistance rates to clindamycin were observed, especially for Gram-positive anaerobic cocci (GPACs) (54.5%) and for *Bacteroides* spp. (45.1%). Overall, low resistance rates to almost all antibiotics were observed for *Clostridium*. High resistance rates to penicillin were also observed for Gram-positive anaerobic bacilli (GPABs) (44.4%), as well as to metronidazole (22.2%), although only nine isolates were included. Antimicrobial susceptibility testing for anaerobes should always be performed in severe infections, such as those localized in the bloodstream. The information obtained contributes to selecting empirical treatments according with local data on resistance.

## 1. Introduction

Anaerobic bacteria are common pathogens in humans. The majority of anaerobes are involved in mild mixed infections, but, in some circumstances, they may cause severe diseases, like when present in blood or in other sterile body sites. The presence of anaerobic microorganisms in the bloodstream continues to have a high associated mortality rate requiring a rapid diagnostic and an appropriate treatment [1,2]. The detection rate of anaerobes in blood cultures is around 0.5–11% of all bacteraemic episodes, depending on patient age and condition and on geographic location [3]. 

Antimicrobial susceptibility testing of anaerobic bacteria is sometimes difficult, and it is performed by few laboratories; only 19% of laboratories in Scotland and 21% of laboratories in the USA carry out anaerobe identification of antimicrobial susceptibility [4,5,6]. Antimicrobial susceptibility testing of anaerobic isolates is currently needed for severe infections (such as bloodstream infections) or for epidemiological studies [7,8], but the observation of higher levels in the resistance of anaerobes to some antimicrobial agents over the last years emphasizes the need for this testing in many more situations [7]. However, care must be taken in handling these kinds of microorganisms due to critical factors for successful isolation and antimicrobial testing, such as correct incubation in an anaerobic atmosphere, the use of specialized culture media, and prolonged culture [9].

Reviewing the medical scientific literature, there are few data on the resistance of anaerobes worldwide, so more efforts should be applied to significantly increase the number of studies in this field. Here, we report the percentage of resistance of clinically significant anaerobes isolated from bloodstream infections in a tertiary-care hospital in Spain. Moreover, the main clinical features of these patients have been also recorded.

## 2. Results

### 2.1. Characteristics of Patients

The study finally included 141 cases of bacteraemia from 141 patients, 55% (*n* = 78) males, with a mean age of 65 years (ranging from 16–98). Fifteen cases of bacteraemia caused by anaerobic pathogens were excluded because they were produced by *Cutibacterium acnes* (formerly *Propionibacterium acnes)* and isolated only from one set of blood cultures. The overall incidence of anaerobic bacteraemia over this period was 1.32 episodes/1000 admissions. One hundred and twenty-one (85.1%) of the episodes were community-acquired and 20 (14.9%) were hospital-acquired. As all episodes were considered clinically relevant, all patients were treated with antimicrobials, except for one patient for whom no records were obtained. One hundred and nine (77.3%) patients were treated with only one antibiotic, and 31 (21.9%) patients were treated with two or more antibiotics. The most used antimicrobial was amoxicillin-clavulanic acid, followed by piperacillin-tazobactam, metronidazole, and meropenem. Clindamycin was used in only 5 episodes. Most of the patients had serious underlying diseases or conditions at the time of the bacteraemia, especially cancer, surgical procedures, and treatment with antimicrobials and/or corticosteroids. 

Table 1 exhibit the clinical characteristics of the 141 patients included in this study.

### 2.2. Isolated Bacteria

Table 2 shows the anaerobic bacteria isolated over the period of study. One hundred and forty-one clinically relevant anaerobic strains causing bloodstream infection were included: 53.1% (*n* = 75) were Gram-negative bacilli, with 43.9% (*n* = 62) being genus *Bacteroides*, 7% (*n* = 10) genus *Fusobacterium*, and 2.1% (*n* = 3) genus *Prevotella*; 30.4% (*n* = 43) were Gram-positive bacilli, with 24.1% (*n* = 34) being *Clostridium* and 2.8% (*n* = 4) *Egghertella lenta*. On the other hand, Gram-positive anaerobic cocci (GPACs) represented 15.6% (*n* = 22) of all isolates, most frequently *Peptoniphilus* (4.2%, *n* = 6) and *Parvimonas micra* and *Anaerococcus* spp. (3.5%, *n* = 5, each one). Moreover, four (2.8%) isolates of *Finegoldia magna* were also cultured. Only one (0.7%) strain of Gram-negative anaerobic cocci was isolated (*Veillonella parvula*). Overall, in 75% of all bacteraemias, *Bacteroides* spp., *Clostridium* spp., and *Fusobacterium* spp. were isolated. 

### 2.3. Antimicrobial Susceptibility

Figure 1 exhibits the antimicrobial susceptibility results for the most important anaerobic bacteria causing bloodstream infection. Numerical percentages have been provided as Appendix A. The most frequently isolated *Bacteroides* spp. were *B. fragilis* (*n* = 42), *B. thetaiotaomicron* (*n* = 8), and *B. vulgatus* (*n* = 6). Resistance to benzylpenicillin was found for 98.3% of *Bacteroides* spp. isolates, resistance to clindamycin for 45.1%, and resistance to moxifloxacin for 35.4%. Only 3.2% of isolates were resistant to imipenem. Two rare isolates of *Bacteroides* were resistant to metronidazole (*B. ovatus*, MIC > 256 µg/mL; *B. thetaiotaomicron*, MIC 6 µg/mL). 

Among 34 isolates of *Clostridium*, *C. perfringens* was the most common microorganism in this genus (*n* = 23). Overall, resistance to penicillin was observed for 8.8%, resistance to metronidazole for 11.7%, and resistance to clindamycin for 2.5%. All strains were susceptible to imipenem and the majority of them to amoxicillin-clavulanic acid. Regarding other Gram-positive anaerobic bacilli (GPABs), 44.4% were resistant to benzylpenicillin, and 22.2% were resistant to both metronidazole and moxifloxacin. No resistance in this group was observed to imipenem and amoxicillin-clavulanic acid. Finally, among Gram-positive anaerobic cocci (GPACs, *n* = 22), resistance to benzylpenicillin was found for 4.7%. High resistance rates to clindamycin were found for this group (54.5%), as well as for moxifloxacin (23.8%). No resistance was observed to imipenem and amoxicillin-clavulanic acid. 

Only 20 isolates were from community-acquired origins. There were no differences in antimicrobial susceptibility between strains of community-acquired and nosocomial-acquired origin.

## 3. Discussion

This study is focused on the clinical characteristics and antimicrobial susceptibility of 141 clinically relevant anaerobic bacteria isolated in the routine testing of bloodstream infections in a tertiary-care hospital in Spain. The main anaerobic microorganisms isolated here belonged to the genera *Bacteroides* and *Clostridium* (see Table 2). Blood test markers, especially C-reactive protein levels, were elevated in 124 cases (87.9%). The main risk factor for bloodstream anaerobic infection in the present report was the presence of cancer (56/39.37%). The majority of patients included here had a favorable outcome, although the associated mortality rate was 24.8% (35 patients deceased). However, the attributable mortality rate to anaerobic bacteraemia was 20% (29 from 141 patients). 

*Bacteroides* (*n* = 62) showed a high rate of resistance to clindamycin (45.1%), moxifloxacin (35.4%), and amoxicillin-clavulanic acid (32.2%). However, a low rate of resistance to metronidazole and imipenem was observed (3.2% each one). Of course, resistance to penicillin was observed in 98.3% of the isolates tested, similar to the rate published in other studies [10,11,12]. In these studies, and in others [13,14], the resistance rate of *Bacteroides* to clindamycin is high and similar to our study. However, a report (*n* = 13) showed the highest resistance was to clindamycin in previous years (61.5%) [15]. Regarding moxifloxacin, all mentioned studies showed less resistance of *Bacteroides* to this drug [10,11,12,13], except two studies that showed a similar rate [14,15]. Overall, resistance of *Bacteroides* to metronidazole remains rare [10,12,13], although in two studies some cases of metronidazole resistance could be observed, as in our case [11,12]. In our series, only two strains of *Bacteroides* (*B. ovatus* and B. *thetaiotaomicron*) isolated from bloodstream infection showed resistance to metronidazole; this resistance has not been observed for all species of *Bacteroides* [16]. In a very recent study, only one isolate of *B. fragilis* was resistant to metronidazole [14]. Low rates of resistance to imipenem were also observed in these studies [11,12,13,14,15], whereas, in the study of Wang et al., the average of resistance of *Bacteroides* to this drug was 8.1% [10]. 

In our series, very few isolates of *Fusobacterium* (*n* = 10) were resistant to antimicrobials. In our study, *Fusobacterium* spp. only showed low resistance rates to penicillin and moxifloxacin (10% for each one) and greater resistance to clindamycin (20%). A study found higher resistance rates than in ours to clindamycin (33%), imipenem (22%), moxifloxacin (22%), and piperacillin-tazobactam (11%) in 16 isolates [12]. On the other hand, another work showed higher resistance rates of *Fusobacterium* to moxifloxacin (44%), clindamycin (33.3%), and imipenem (22.2%) [11]. A study with increased isolation of *Fusobacterium* in bloodstream infections showed low resistance rates to almost all antibiotics (except penicillin) in *F. nucleatum* (29 isolates) but higher resistance rates in other species than *F. nucleatum* (including penicillin) (19 isolates) [10].

Among *Clostridium* spp. (*n* = 34 isolates), resistance was observed for all antimicrobials tested except for imipenem. However, the resistance rate for the strains was low or very low (highest for metronidazole, 11.7%). Very low resistance rates were found for the six strains of *Clostridium perfringens* in the study of Umemura et al. [14]. However, in a recent study in Singapore with 34 *Clostridium* isolates, higher resistance rates of *Clostridium* were observed for all antimicrobials tested than in our series [12], especially for clindamycin (35%), moxifloxacin (33%), metronidazole (19%), and penicillin (19%). The results of the report from Taiwan are also noteworthy [10]: from 50 strains of *C. perfringens*, only 6% resistance to clindamycin was detected without any detected resistance to other antimicrobials, but, from 43 isolates of other *Clostridium* species, higher resistance rates were found, especially to clindamycin (27.9%), penicillin (16.3%), and moxifloxacin (14%). 

Fifteen percent of the isolates in our series were GPACs (*n* = 22) and they had a different resistance profile to that of the Gram-negative anaerobes. The most striking in this group is the high resistance to clindamycin (54.5%), followed by moxifloxacin (22.7%). Only one strain from this group was resistant to penicillin, metronidazole, or piperacillin-tazobactam. No resistance to imipenem and amoxicillin-clavulanic acid was observed in any genera. There is not much data about the resistance of GPACs to antimicrobials isolated from blood cultures in the medical literature. The study by Wang et al. [10] with 20 strains of *Parvimonas micra* (formerly *Peptostreptococcus micros*) and 20 isolates of other *Peptostreptococcus* species showed only 16.7% resistance to clindamycin and 6.7% resistance to metronidazole in *P. micra*; other *Peptostreptococcus* species showed 20% resistance to clindamycin and 15% resistance to penicillin. All these bacteria were susceptible to the remaining antimicrobials. 

The last group referred to other Gram-positive anaerobic bacilli and showed a high resistance rate to penicillin (44.4%), metronidazole (22.2%), and moxifloxacin (22.2%). As in GPACs, no resistance was obtained to imipenem and amoxicillin-clavulanic acid.

The main limitation of this study was the small number of isolates of some genera, such as *Veillonella*, *Prevotella*, other GPABs, and *Fusobacterium*. This fact prevents the drawing of any conclusions on the antimicrobial susceptibility of these anaerobic bacteria. However, this fact is a common finding in all studies since these bacteria are infrequently causing bacteremia. Another limitation of the study is that, due to the retrospective design, the agar dilution reference method was not used. This method cannot be used in routine practice, and most laboratories use alternative techniques, such as the Etest method. This fact may induce some variations in antimicrobial resistance rates between studies.

## 4. Conclusions

This study summarizes the findings of resistance to some antimicrobials in 141 strains of anaerobic bacteria isolated from bloodstream infections. Overall, resistance to imipenem and amoxicillin-clavulanic acid is zero or very low in most groups of pathogens. On the other hand, clindamycin had the highest resistance rates in almost all anaerobic microorganisms, so these results confirm that clindamycin should not be used as empirical monotherapy against anaerobes. High resistance rates to penicillin were also observed in the majority of anaerobes. However, it should be highlighted that the majority of isolates in this study were *Bacteroides* spp., and these microorganisms are intrinsically resistant to penicillin. Routine antimicrobial susceptibility testing for anaerobic bacteria should be mandatory, especially in bloodstream infections. Antimicrobial testing contributes to global analysis of the patterns of resistance and allows empirical treatment to be selected according with local or regional data.

## 5. Materials and Methods

### 5.1. Patients

The clinical records of all patients with positive blood cultures for anaerobic microorganisms at the University Hospital Virgen de las Nieves (Granada, Spain) in the period from January 2016 to April 2020 were reviewed retrospectively. This hospital is an approximately 700-bed tertiary-care institution, serving as a primary facility for nearly 500,000 inhabitants. Blood cultures were taken at the request of the attending physicians, who made the decisions concerning the patients’ diagnosis and treatment. Clinically relevant episodes were considered when the patient had one or more positive blood cultures and met one of the following criteria: white blood cell count <4000 or >12,000/µL; temperature >38 °C; or physical, pathological, or surgical evidence consistent with infection [15].

The study was conducted in accordance with the Declaration of Helsinki, and the protocol was approved by the Ethics Committee of our hospital. 

Data were gathered on age and sex, type of anaerobic bacterium, presence of fever, type of infection (nosocomial or community-acquired), risk factors or underlying diseases, laboratory findings, previous surgery, presence of any type of cancer and/or cytotoxic treatment, previous treatment with antimicrobials and corticosteroids, and presence of coagulopathy. Moreover, data on clinical manifestations, treatment, and outcome were also recovered. Repeated isolates from the same patients were excluded as well as the polymicrobial blood cultures.

### 5.2. Clinical Definitions

Bacteraemia or bloodstream infection was defined as nosocomial if positive blood cultures were obtained after 48 h or more had elapsed since hospital admission. On the other hand, bacteraemia was considered to be community-acquired when anaerobes were isolated from blood cultures taken within 48 h of hospital admission and the patient had not been hospitalized in the previous 2 weeks.

The following predisposing factors for bacteraemia were considered: (i) surgical procedures requiring general anaesthesia or cytotoxic agents within the previous month; (ii) use of antibiotics or corticosteroids within 10 days before the episode of bacteraemia; (iii) presence of cancer, diabetes mellitus, chronic diseases, and transplantation.

All these clinical variables were recorded for the day on which the initial positive blood culture was obtained.

### 5.3. Microbiology: Isolation and Identification of Strains

Two blood culture sets were usually drawn from an antecubital vein. Blood samples were inoculated into both aerobic and anaerobic blood culture bottles and incubated in the BD BACTECTM FX system (BD, Becton Dickinson) for a maximum of 5 days. Positive anaerobic bottles were inoculated onto aerobic and anaerobic blood agar (BD Columbia Agar 5% Sheep Blood, Becton Dickinson, Franklin Lakes, NJ, USA), incubating the plates at 35–37 °C for a maximum of 5 days. Anaerobic plates were incubated in an anaerobic atmosphere generated with the AnaeroGen Compact anaerobic system (Oxoid Ltd, Wide Road, Basingstoke, England) at 35–37 °C. 

Identification of all isolates was carried out using MALDI-TOF MS (Bruker Biotyper, Bellerica, MA, USA), following the manufacturer’s recommendations. Only strains clinically relevant and with a log (score) ≥ 2.0 were included and interpreted with high confidence [17]. Some isolates (*n* = 5) were also identified by 16S rRNA gene sequencing, when the MALDI-TOF MS score was lower than 2.0.

### 5.4. Antimicrobial Susceptibility Testing

Antimicrobial susceptibility testing of isolates was performed with the gradient diffusion method using Etests (bioMérieux, Marcy l’Etoile, France) against seven antibiotics: benzyl-penicillin, amoxicillin-clavulanic acid, piperacillin-tazobactam, imipenem, moxifloxacin, clindamycin, and metronidazole. The method was performed according to CLSI Standard M11 A8 [18]. All anaerobic strains were sub-cultured in *Brucella* agar supplemented with 5% laked sheep blood, hemin, and 10 µg/mL vitamin K1 (BD, Becton Dickinson), and plates were incubated in an anaerobic atmosphere at 35–37 °C for 48 h. Antimicrobial susceptibility results were interpreted as “susceptible”, “resistant” or “susceptible or increased exposure” (formerly “intermediate”) according to EUCAST breakpoints, except for moxifloxacin (CLSI breakpoints were used in the absence of EUCAST breakpoints) [19,20]. *B. fragilis* ATCC 25285, *C. perfringens* ATCC 13124, and *P. anaerobius* ATCC 27377 were used for monthly quality control tests. 

## Figures and Tables

**Figure 1 antibiotics-09-00345-f001:**
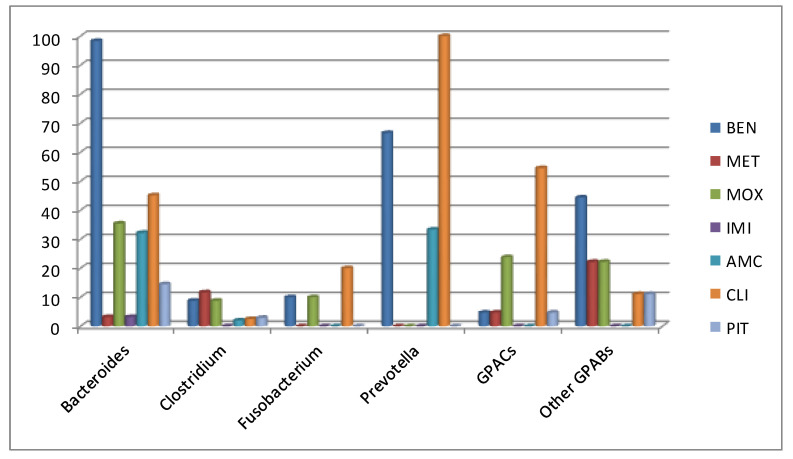
Resistance rate (%) of anaerobic bacteria against selected antimicrobial agents obtained from bacteraemias. BEN: benzylpenicillin; MET: metronidazole; MOX: moxifloxacin; IMI: imipenem; MER: meropenem; AMC: amoxicillin-clavulanate; CLI: clindamycin; PIT: piperacillin-tazobactam; GPACs: Gram-positive anaerobic cocci; GPABs: Gram-positive anaerobic bacilli.

**Table 1 antibiotics-09-00345-t001:** Characteristics of 141 patients with bloodstream infection caused by anaerobic microorganisms.

Characteristics	*n* (%)
**Clinical findings ***	
Fever	108 (76.5)
**Digestive symptoms (abdominal pain, diarrhea, vomiting, etc.)**	54 (38.2)
Respiratory symptoms (chest pain, cough, dyspnea, etc.)	18 (12.7)
Neurological symptoms (paralysis, loss of conscience, etc.)	11 (7.8)
General symptoms (malaise, asthenia, etc.)	17 (12)
**Laboratory results**	
Increased CRP level	124 (87.9)
Increased Pct level	32 (66.6) **
Increased WCBC level	71 (50.3)
**Underlying diseases or conditions ***	
Cancer	54 (39.7)
Intestinal or abdominal	22 (15.6)
Hematologic	17 (12)
Other	17 (12)
Surgery	41 (29)
Transplantation	8 (5.6)
Cytotoxic therapy	31 (21.9)
Diabetes mellitus	33 (23.4)
Chronic diseases	29 (20.5)
CT and Atb treatment	77 (54.6)
**Treatment**	
One drug	109 (77.3)
More than one drug	31 (21.9)
**Outcome**	
Favorable	106 (75.1)
Deceased	35 (24.8)

***** Some patients had more than one symptom and/or underlying condition * Percentage calculated from 48 patients who were tested. CRP: C-reactive protein; Pct: procalcitonin; WCBC: white cell blood count; CT: corticosteroids; Atb: antibiotics.

**Table 2 antibiotics-09-00345-t002:** Anaerobic microorganisms isolated from bloodstream infections.

Microorganisms	*n*	%
**Gram-negative bacilli**	**75**	**53.1**
*Bacteroides*	62	43.9
*B. fragilis*	42	29.7
*B. thetaiotaomicron*	8	5.6
*B. vulgatus*	6	4.2
*B. uniformis*	3	2.1
*B. ovatus*	3	2.1
*Prevotella*	3	2.1
*P. buccae*	1	0.7
*P. baroniae*	1	0.7
*P. intermedia*	1	0.7
*Fusobacterium*	10	7
*F. nucleatum*	9	6.3
*F. necrophorum*	1	0.7
**Gram-negative cocci**	**1**	**0.7**
*Veilonella parvula*	1	0.7
**Gram-positive bacilli**	**43**	**30.4**
*Clostridium*	34	24.1
*C. perfringens*	23	16.3
*C. clostridioforme*	3	2.1
*C. septicum*	2	1.4
*C. baratii*	1	0.7
*C. butyricum*	1	0.7
*C. ramosum*	3	2.1
*C. sordellii*	1	0.7
*Eggerthella lenta*	4	2.8
*Propionibacterium lymphophylum*	2	1.4
*Eggerthia catenaformis*	2	1.4
*Eubacterium limosum*	1	0.7
**Gram-positive cocci**	**22**	**15.6**
*Finegoldia magna*	4	2.8
*Peptoniphilus*	6	4.2
*P. harei*	5	3.5
*P. gorbachii*	1	0.7
*Parvimonas micra*	5	3.5
*Peptostreptococcus anaerobius*	1	0.7
*Anaerococcus*	5	3.5
*Anaerococcus spp*	2	1.4
*A. lactolyticus*	1	0.7
*A. tetradius*	1	0.7
*A. prevotii*	1	0.7
*Peptococcus niger*	1	0.7
**Total**	**141**	**100**

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
