# Peer review of "Clinical Findings and Antimicrobial Susceptibility of Anaerobic Bacteria Isolated in Bloodstream Infections"

_antibiotics, 2020, doi:10.3390/antibiotics9060345_

Round 1

Reviewer 1 Report

Dear Authors,

This study examined the drug-resistance of anaerobic bacteria isolated from patients who fell into bacteremia. I think this study can well understand the situation of the prevalence of drug-resistance of anaerobic bacterial species that are causes of bloodstream infection. The data obtained from this study will serve to help physicians who attempt chemotherapy against the bacteremia caused by anaerobic bacteria in Granada.

Did the susceptibility of anaerobic bacterial species to antibiotics differ between the bacterial strains obtained from nosocomial-infected patients and community-acquired patients?

Please carefully check grammar, binominal nomenclatures (are they written by Italic?), and misspells.

Sincerely yours,

Author Response

1- Did the susceptibility of anaerobic bacterial species to antibiotics differ between the bacterial strains obtained from nosocomial-infected patients and community-acquired patients?

R- Only 20 strains were from community-acquired origin. There were no substantial differences between both kind of strains. We have added this statement in lines 111-112.

2- Please carefully check grammar, binominal nomenclatures (are they written by Italic?) and misspells.

R- We have check the grammar, the binomial nomenclatures and the misspells.

Reviewer 2 Report

Cobo et al. describes clinically determined bacteremia in 141 human subjects. Blood cultures were used and species identification was done by primarily mass spectrometry (MALDI-TOF) and supplemented with the 16s ribosomal RNA sequencing. The study is important and experimental details are good. However, article suffers from not so good representation of work.

I have following concerns that authors must address:

  1. There are no figures in paper, it will good to have some key figures for blood cultures, images of culture plates from some sample patients.
  2. Primary data for the antimicrobial susceptibility tests should be plotted. All data in its numeric form needs to be provided as supplementary datasets, as that can serve as a resource.
  3. Wherever a claim is made try to put some statistics (pvalues) should be indicated, at least for the claims made in the abstract. Authors must stress with confidence if observation is in-line with expectation or they are novel.
  4. Table 3 would be much better if it is plotted instead of displayed as a table.

Author Response

1- There are no figures in paper, it will good to have some key figures for blood cultures, images of culture plates from some sample patients.

R- We acknowledge to the reviewer this statement but, although we can add these figures, we consider that no important/relevant information add to this paper. Moreover, there is a great variety of anaerobic microorganisms included in this study in order to choose photographs of the cultures.

2- Primary data for the antimicrobial susceptibility tests should be plotted. Table 3 would be much better if it is plotted instead of displayed as a table.

R- We have plotted these data and we have provided numerical data as supplementary data.

3- Wherever a claim is made try to put some statistics (pvalues) should be indicated, at least for the claims made in the abstract. Authors must stress with confidence if observation is in-line with expectation or they are novel.

R- We think that we have put all statistics results from the most important data through the manuscript. We have no put p values because no statistical study was performed.

Reviewer 3 Report

General comments

This manuscript provides an interesting overview of the current antimicrobial resistance rate of anaerobic bacteria isolated in blood stream infections in Spain. To improve the readability of the manuscript and interpretation of the results, some modifications are suggested.

Specific comments

Abstract.

Line 16. The MALDI-TOD MS system should be given (Bruker).

Line 19. Replace “(except for moxifloxacin)” by “except for moxifloxacin (CLSI breakpoints).

Line 21-22: “Almost…”. This sentence suggests that many Bacteroides and Prevotella isolates were resistant to imipenem and amoxiclav which does not fit presented data. Considering that this study included only 3 Prevotella isolates, this sentence should be rephrased, and authors should only provide Bacteroides resistance rates.

Line 22-23. “and for genera Bacteroides”: replace by “and for Bacteroides spp”.

Line 24. Gram positive anaerobic bacilli. Authors should first report resistance rates of Clostridium spp, then of other GPAB. In the abstract, the resistance rates of “GPACs” seems to refer to “GPABs” (typo error). Considering that only 9 non Clostridium- GPAB isolates were identified, reporting a 22% metronidazole resistance rate is misleading (only 2 isolates in fact). Authors should provide information on the number of isolates.

Line 26. English. The last sentence is unclear and should be rephrased.

Methods

  • Line 202. Authors should precise the criteria they used to select “clinically relevant” anaerobes (see line 79).
  • Authors should provide evidence that they obtained the agreement of an ethical committee to perform this study.

Results

  • The total number of patients with positive anaerobic blood cultures (i.e. before exclusion of clinically non-significant isolates) should be given at the beginning of the result section
  • Identification results of isolates that were excluded for this study should be given.
  • Did authors fail to identify some isolates?
  • Line 78. Anaerobes associated with bloodstream infections, after exclusion of non clinically relevant isolates
  • Lines 78-88. Bacterial genera or species are not italicized

Discussion

General comments.

In the discussion section, authors compare the antimicrobial resistance rates of Bacteroides, Fusobacterium, Clostridium spp and GPAC observed in their study to other reports. For a better understanding / interpretation, author should both provide information on the number of studied isolates in their study and others, and where and when other studies were performed. In addition, AST methods and breakpoints should be given. Authors could add these data in table 3, or provide a supplementary figure.

Specific comments.

  • Line 123. If feasible, authors should precise the % of deaths attributable to anaerobic bacteremia or to the underlying condition.
  • Line 141. “Fusobacteria” refers to a phylum, not to a genus. The correct wording is “Fusobacteriumspp”.
  • Line 145. The reference is missing.
  • Line 162. Typographical error (“ppiperacillin-tazobactam”).
  • Limitations of the study. Another limitation of the study is that, due to the retrospective design of the study, Authors did not use the agar dilution reference method. A critical issue of AST of anaerobes is that this reference method cannot be used in routine practice, and that most laboratories use alternative techniques (such as the e-test technique). This may induce variations of antimicrobial resistance rate between studies, this should be also reported.

Conclusion

Line 181. Note: These results confirm that clindamycin should not be used as empirical monotherapy against anaerobes.

Line 182. Authors state that resistance rates to penicillin where high in a majority of anaerobes. Considering that most anaerobic isolates were Bacteroides spp which are intrinsically resistant to penicillin, authors should provide a more nuanced picture.

Line 184. “This performance … “ The sentence is unclear and should be rephrased.

Figure 3 legend.

Authors should provide the meaning of GPAS.

Author Response

Abstract

1- Line 16- The MALDI-TOF MS system should be given (Bruker).

R- We have added this information.

2- Line 19- Replace “except for moxifloxacin” by except for moxifloxacin (CLSI breakpoints).

R- We have replaced it, as requested.

3- Line 21-22- “Almost…..”.

R- We agree with the reviewer. We have deleted Prevotella.

4- Line 22-23: “and for genera Bacteroides”. Replace by “and for Bacteroides spp”.

R- We have replaced it.

5- Line 24: “Gram positive anaerobic bacilli…………..Authors should provide information on the number of isolates”.

R- We have put the number of isolates, we have clearified the mean of GPACs and GPABs, we have put firstly the Clostridium results.

6- Line 26. English. The last sentence is unclear and should be rephrased.

R- We have rephrased the last sentence.

Methods

1- Line 202. Authors should precise the criteria they used to select “clinically relevant” anaerobes (see line 79).

R- We have added these criteria in the text.

Authors should provide evidence that they obtained the agreement of an ethical committee to perform this study

R- In our hospital, all studies are studied and approved by the ethical committee. The present study was also approved.  

Results

1- The total number of patients with positive anaerobic blood cultures should be given at the beginning of the result section.

R- We have added the patients excluded from this study and the reason.

2- Identification results of isolates that were excluded for this study should be given.

R- All isolates excluded from this work were identified as Cutibacterium acnes, and were isolated only from one set of blood cultures, being considered as probable contaminants.

3- Did authors fail to identify some isolates?

R- Some isolates (n=5) were not identified by MALDI-TOF MS, but they were identified by 16S rRNA gene sequencing, when MALDI-TOF MS score was lowest than 2.0.

4- Lines 78-88- Bacterial genera or species are not italicized.

R- We are very sorry, now we have italicized all the names of bacteria genera and species.

Discussion

1- In the discussion section, authors compare the antimicrobial resistance rates……….authors should provide information on the number of the studied isolates in their study and others and where and when other studies were performed.

R- We have added the number of isolates in our study and in the remaining studies.

2- AST methods and breakpoints should be given.

R- AST methods are given in the Antimicrobial susceptibility testing section (Lines 244-254). Breakpoints used were those from EUCAST (reference number 19) except from moxifloxacin (reference number 18).

3- Line 123- If feasible, authors should precise the % of deaths attributable to anaerobic bacteraemia or to the underlying condition.

R- In the line 134, we have added the attributable mortality to anaerobic bacteraemia in this study.

4- Line 141- Fusobacteria refers to a phylum, not to a genus. The correct wording is “Fusobacterium spp”.

R- Of course. Thank you. We have changed it.

5- Line 145. The reference is missing.

R- Yes, we have added the reference (number 11).

6- Line 162- Typographical error

R- We have corrected it.

7- Limitations of the study.

R- We have added these limitations friendly provided by the reviewer.

Conclusion

1- Line 181. Note: These results confirm that clindamycin should not be used as empirical monotherapy against anaerobes.

R- Yes, of course. We have added this statement in the text.

2- Line 182- Authors state that resistance rates to penicillin where high in a majority of anaerobes. Considering that most isolates were Bacteroides spp which are intrinsically resistant to penicillin, authors should provide a more nuanced picture.

R- We agree with the reviewer. We have added a statement regarding to this in this section.

3- Line 184. “This performance……”. The sentence is unclear and should be rephrased.

R- We have rephrased this statement.

Figure 3 legends.

1- Authors should provide the meaning of GPACs (and GPABs)

R- We have provided it in the abstract and in the tables/figures.

Round 2

Reviewer 2 Report

Manuscript is much better now and authors provided appropriate responses.